# Optimization of Culture Media and Feeding Strategy for High Titer Production of an Adenoviral Vector in HEK 293 Fed-Batch Culture

**DOI:** 10.3390/vaccines12050524

**Published:** 2024-05-10

**Authors:** Chun Fang Shen, Anja Rodenbrock, Stephane Lanthier, Elodie Burney, Martin Loignon

**Affiliations:** Human Health Therapeutics Research Centre, National Research Council of Canada, Montreal, QC H4P 2R2, Canada

**Keywords:** adenovirus, HEK 293 cells, fed-batch culture, feeding strategy

## Abstract

Adenoviruses are efficient and safe vectors for delivering target antigens and adenovirus-based vaccines have been used against a wide variety of pathogens, including tuberculosis and COVID-19. Cost-effective and scalable biomanufacturing processes are critical for the commercialization of adenovirus-vectored vaccines. Adenoviral vectors are commonly produced through the infection of batch cultures at low cell density cultures, mostly because infections at high cell densities result in reduced cell-specific virus productivity and does not improve volumetric productivity. In this study, we have investigated the feasibility of improving the volumetric productivity by infecting fed-batch cultures at high cell densities. Four commercial and one in-house developed serum-free media were first tested for supporting growth of HEK 293 cells and production of adenovirus type 5 (Ad5) in batch culture. Two best media were then selected for development of fed-batch culture to improve cell growth and virus productivity. A maximum viable cell density up to 16 × 10^6^ cells/mL was achieved in shake flask fed-batch cultures using the selected media and commercial or in-house developed feeds. The volumetric virus productivity was improved by up to six folds, reaching 3.0 × 10^10^ total viral particles/mL in the fed-batch culture cultivated with the media and feeds developed in house and infected at a cell density of 5 × 10^6^ cells/mL. Additional rounds of optimization of media and feed were required to maintain the improved titer when the fed-batch culture was scaled up in a bench scale (3 L) bioreactor. Overall, the results suggested that fed-batch culture is a simple and feasible process to significantly improve the volumetric productivity of Ad5 through optimization and balance of nutrients in culture media and feeds.

## 1. Introduction

Adenovirus (Ad)-based vectors are highly efficient gene transfer vectors widely used in vaccine development [1,2] and a variety of gene therapy applications [3]. The Ad-based vaccine platform represents an attractive strategy as it induces robust humoral and cell-mediated immune responses, with proven safety and vaccine efficacy, and can meet the global demand in a pandemic situation. The development of at least four Ad vector-based COVID-19 vaccines, their excellent protection profiles and the administration of over one billion doses so far have fully elucidated the potential of this vaccine delivery system.

When adenovirus is used for vaccine applications, the price per dose will be an important determining factor of its economic viability. Achieving a cost-effective and scalable manufacturing process will be critical for success [4], particularly when the diseases targeted by Ad vector-based vaccines have their highest prevalence in resource-poor settings. Therefore, the biomanufacturing process must yield high Ad titers. Currently, the Ad production process used at either the R&D setting or commercial manufacturing is still a typical batch culture process with an optimal cell density between 1 to 2 × 10^6^ cells/mL at infection. There is a potential to improve the process productivity by infecting cultures at higher densities; however, many have attempted to improve the volumetric productivity using this strategy, but have not achieved much success [5,6,7]. Adenovirus production is limited by reduced cell-specific productivity when high cell density cultures are infected. The breakpoint related to specific production drop with increasing cell densities at infection depends on the cell culture media. This limitation has been referred to as the “cell density effect” [8].

The cell density effect has been generally associated with nutrient limitations and/or accumulation of inhibitory metabolites, although the exact nature of these limitations remains largely unknown [9]. A complete medium replacement at the time of infection has been commonly used as a strategy to reduce nutrient limitations and/or the accumulation of inhibitory metabolites in batch operations. Virus production was significantly higher when cultures were resuspended in the fresh medium compared to those without medium replacement at similar infected cell densities [10]. Higher volumetric productivity was achieved and maintained at infected cell densities up to 3 × 10^6^ cells/mL in shake flask culture [7].

Perfusion culture offers a continuous supply of fresh nutrients, maintaining required levels of essential nutrients, while removing inhibitory metabolites from the culture. The perfusion process allows the culture to reach a high cell density and provide the nutrients required during the virus production phase to maintain cell-specific virus productivity or minimize the “cell density effect” in cultures infected at higher cell densities. With the development of alternating tangential flow (ATF) technology, perfusion culture is becoming a popular process for the production of viruses and viral vectors [11]. Cell-specific virus yields could be maintained or the volumetric productivity was proportionally improved with the cell densities up to 7 × 10^6^ cells/mL at infection [12,13,14].

Both batch culture with a medium replacement at infection and perfusion culture require additional equipment, and the use and handling of substantial quantities of additional medium, especially at manufacturing scales. Some perceive it as being a complex operation that is more difficult to implement than a fed-batch process. In addition, media consumption is substantially higher, and the operational cost of perfusion culture is much higher.

Fed-batch culture has become the research focus in the pharmaceutical industry to improve process volumetric productivity and reduce production costs. In the fed-batch process, the initial cellular growth is supported by the basal medium. Concentrated feed is added in fed-batch culture to replenish nutrients, sustain the cell growth to high density and improve productivity [15]. Fed-batch processes are widely used to produce monoclonal antibodies but have only occasionally been employed in the process development for virus production [16,17]. Some previous works [7,18] employed a fed-batch strategy to alleviate the detrimental effects of lactate and ammonia accumulation on virus production, and slightly improved volumetric virus production. The virus productivity was still lower when compared to the medium replacement strategy. Nutrients provided by the relatively simple feeds might not meet the complexity of nutritional requirements in both the cell growth and viral production process phases [19]. Recently, some improvement was achieved in fed-batch culture for the production of a chimpanzee adenovirus-vectored SARS-CoV-2 vaccine [5].

Based on our recent findings that titers of other types of viruses can be increased using commercially available or in-house developed media [20,21], we seek to increase Ad5 titers by re-examining the fed-batch culture process. Through testing and customizing media, feed and regimen, we have succeeded in infecting HEK 293 fed-batch cultures at cell densities up to 5 × 10^6^ cells/mL while maintaining the cell-specific productivity, resulting in 6-fold improvement in the volumetric titers up to 3 L bench scale bioreactor.

## 2. Materials and Methods

### 2.1. Cell Line, Cell Culture Media and Adenoviral Strain

The HEK 293SF cell line was derived from HEK 293S cells, and adapted to grow in suspension and serum-free media through multiple steps of adaptation and clone selection [22]. Cells were grown in suspension in shake flasks (Corning) with an agitation of 120 rpm using orbital shakers (Infors HT, Anjou, QC, Canada), 5% CO_2_ at 37 °C. Cells were passaged regularly when reaching densities ~1 × 10^6^ cells/mL.

The following culture media were utilized in this study: Pro293s-CDM (Lonza, Walkersville, MD, USA), SFM4Transfx-293 and SFM4HEK293 (both from Cytiva, Logan, UT, USA), Ex-Cell 293 (Sigma-Aldrich, St. Louis, MO, USA) and an in-house-developed serum-free medium (HEK SFM).

The adenovirus used for infection was an Ad5-containing green fluorescence protein (Ad5-GFP) under the control of the CMV promoter. Viral stock titer was 1.7 × 10^10^ IVP/mL and aliquots were stored at −80 °C for subsequent analysis.

### 2.2. Batch Culture

A vial of HEK 293SF cells was thawed and maintained in SFM4Transfx-293. The HEK 293SF cells were adapted and maintained in the selected cell culture media in shake flask culture for at least 2 weeks before the cells were used for batch culture experiment for evaluation of cell growth in different culture media.

For evaluation of Ad5 production, HEK 293SF batch culture was seeded at a cell density between 0.20 to 0.30 × 10^6^ cells/mL in different media. When the culture reached a cell density of 1 ± 0.1 × 10^6^ cells/mL (after two days of seeding), the culture was then infected with Ad5-GFP at a multiplicity of infection of 10 infectious viral particles per cell (MOI = 10 ivp/cell was used throughout the study). The infected culture was harvested at 48 h post infection (hpi) and stored at −80 °C for subsequent analysis.

### 2.3. Feeds and Fed-Batch Cultures

The fed-batch culture was conducted using the two best media (SFM4Transfx-293 and HEK SFM) identified during the media evaluation. In the uninfected fed-batch culture process, the cultures, grown in SFM4Transfx-293 medium, were fed with 3%, 5% and 7% (*v*/*v*) Cell Boost 5 (CB5, Cytiva) when the cultures reached respective cell densities of about 3, 5 and 7 × 10^6^ cells/mL. The CB5 at a concentration of 35 g/L is a chemically defined feed. For the cells cultivated in HEK SFM medium, the culture was fed with various individual component/hydrolysate (such as amino acids, growth factor, trace metal, etc.), CB5 or a series of feeds developed in-house (Table 1) when its cell density reached about 3 × 10^6^ cells/mL to examine their effect on the cell growth and maximum cell density. Four in-house developed feeds (Feeds 1 to 4) were prepared according to Table 1. The fed-batch cultures were sampled for cell count, osmolality measurement and concentration analysis of lactate, ammonia, residual glucose and amine acid.

Some fed-batch cultures were infected with Ad5-GFP when the cell densities reached between 2.5 to 5 × 10^6^ cells/mL. The infected cultures were harvested at 48 hpi and stored at −80 °C for subsequent analysis.

### 2.4. Scale-Up of Fed-Batch Culture Processes to 3 L Bench-Scale Bioreactor

Both cell growth and production of Ad5-GFP were separately scaled up to 3.5 L Chemap bioreactor (Mannedorf, Switzerland) with a working volume of 2.7 L under controlled conditions. The bioreactor was inoculated at a cell density of 0.25 × 10^6^ cells/mL in HEK SFM medium. Temperature, agitation and dissolved oxygen (DO) in the bioreactor were controlled at 37 °C, 90 rpm and 40% of air saturation, respectively. The pH was controlled at 7.2 ± 0.1 by using CO_2_ in the gas inlet and base consisting of 5% NaOH and 7.5% NaHCO_3_. The uninfected fed-batch culture was fed with a series of in-house developed feeds (Feeds 1, 2, 3 and 4) according to the developed feeding schedule. The fed-batch culture for the production of Ad5-GFP was fed with Feed 1 and 2, respectively, at the cell density of about 3 and 5 × 10^6^ cells/mL and infected immediately with Ad5-GFP after the second feeding. The infected culture was harvested at 48 hpi and stored at −80 °C for subsequent analysis.

### 2.5. Assessment of the Influence of Cell Status on the Virus Productivity

HEK 293SF cells were grown in 2 L shake flask (with a working culture volume of 700 mL) and in a 3 L bioreactor with HEK SFM medium under the fed-batch condition developed previously. As illustrated in Figure 1, when the culture grew to different cell densities, a portion of the culture was removed from the 2 L shake flask and 3 L bioreactor, transferred to 125 mL shake flask in duplicate and infected with the virus. Another portion of the culture was centrifuged at 300× *g* for 5 min, and the cell pellet was resuspended in fresh HEK SFM medium at 1 × 10^6^ cells/mL before the viral infection. This procedure ensured the same culture condition was applied to all cells during the production phase and allowed us to examine the effect of non-culture-condition-related factor(s), such as cell status, on productivity.

### 2.6. Analytical Methods

The sample from the shake flask and bioreactor cultures was mixed with an equal volume of Accumax solution (Innovative Cell Technologies, Inc., San Diego, CA, USA) in a 1.5 mL vial. The mixture was then incubated at 37 °C and under mild agitation for 30 min. Cell counts were performed with a Cedex Automated Cell Counter (Roche) or using a hemacytometer and erythrosine B. Concentration of glucose, lactate, ammonia and LDH in culture were quantified by Cedex Bio Analyzer (Roche CustomBiotech, Penzberg, Germany). A HPLC method [19] was used to quantify amino acids in fresh and spent media, and total virus particle titers in harvested culture.

## 3. Results

*Evaluation of culture media supporting HEK 293SF cell growth and Ad5-GFP production in shake flask batch culture: *Figure 2A depicts the cell growth profile of the HEK 293SF batch cultures cultivated in four commercial serum-free media and one in-house developed medium (HEK SFM). Ex-Cell 293 medium supported cell growth close to 6 × 10^6^ cells/mL, while the maximum viable cell densities supported by the other media were lower; up to 4 × 10^6^ cells/mL in HEK SFM and SFM4Transfx-293; and ≤3 × 10^6^ cells/mL in SFM4HEK293 and Pro293s-CDM. The cells’ doubling time in the latter was longer at 40 h.

Figure 2B shows the volumetric Ad5-GFP titers in the batch cultures cultivated with the 5 selected and in-house (HEK SFM) cell culture media, and all were infected at a cell density of 1 ± 0.1 × 10^6^ cells/mL. The cells produced the highest Ad5-GFP titer at 7.5 × 10^9^ vp/mL in SFM4HEK293, slightly higher than titers obtained in HEK SFM at 6.2 × 10^9^ vp/mL. The productivity was between 4.1 to 4.5 × 10^9^ vp/mL in the batch cultures cultivated in Ex-Cell 293 and SFM4Transfx-293. Lastly, Pro293s-CDM was a poor medium in supporting the production of Ad5-GFP.

By taking into consideration the capacity of the cell culture media to support cell growth and virus production, and the availability of commercial and in-house developed feeds, we selected SFM4Transfx-293 and HEK SFM as basal media for the development of a high cell density fed-batch culture process to improve the volumetric productivity of Ad5-GFP.

*Cell growth and Ad5-GFP production in HEK 293SF shake flask fed-batch culture using commercial medium and feed: *Figure 3A depicts very strong growth of HEK 293SF cells cultivated in SFM4Transfx-293 medium and fed with CB5. The total cell density (TCD) reached 18.8 × 10^6^ cells/mL, and viable cell density (VCD) was more than 15 × 10^6^ cells/mL after the culture was fed with a total 15% culture volume of CB5, tripling the maximum cell density obtained in the batch culture. However, the volumetric productivity in the fed-batch culture infected at 2.9 and 4.0 × 10^6^ cells/mL reduced drastically instead of increasing (Figure 3B), indicating that the SFM4Transfx-293—CB5 fed-batch culture process was very promising for reaching a high cell density but did not result in increased Ad5-GFP volumetric titers. This result prompted us to develop in-house feeds to improve the productivity of Ad5-GFP in the fed-batch culture.

*Development of in-house feeds to improve the cell growth and Ad5-GFP production in shake flask fed-batch culture process:* The HEK SFM medium supported the cell growth to 4.3 × 10^6^ cells/mL in batch culture. About a dozen individual nutrients and other supplements such as amino acids, growth factor and trace metal were added alone or in combination to the batch culture when cell cultures reached 3 × 10^6^ cells/mL to further increase cell densities. Experimental data (not shown) only indicated a trivial improvement in the maximum cell density. However, the maximum viable cell density, as shown in Figure 4A, increased to 6.2 × 10^6^ cells/mL and 8.0 × 10^6^ cells/mL when the culture was fed with 1 g/L of yeast extract (Fed-batch_YE) and 1 g/L of Sheff-Vax ACF (Fed-batch_ACF), respectively. Much higher viable cell density up to 15 × 10^6^ cells/mL was obtained when the culture was fed with a series of in-house developed feeds (Feeds 1 to 4; Fed-batch_F1–4) according to the feeding schedule described in Table 1. Surprisingly, the profile of lactate accumulation in the shake flask batch and fed-batch_F1–4 culture was similar even though the maximum viable cell density in the fed-batch culture was almost four times the density achieved in the batch culture Figure 4B. However, the ammonia concentration increased dramatically at day 8 when the cell density reached maximum. The osmolality of the fed-batch_F1–4 culture also increased from 304 to 319 mOsm/kg during the last two days of culture (day 8 to 10).

The volumetric productivity was very promising in the shake flask fed-batch culture using HEK SFM media and in-house developed feeds, and infected at different cell densities. The titer increased almost proportionally to the cell density at infection, reaching 3.0 × 10^10^ vp/mL in the culture infected at 5 × 10^6^ cells/mL, and was 6 times the titer obtained from the batch culture infected at 1.2 × 10^6^ cells/mL (Figure 4C). This promising result prompted us to scale the fed-batch culture process to the bench scale bioreactor to test its scalability.

*Challenges in the scale-up of fed-batch culture to 3 L bioreactor:* The fed-batch conditions developed for the shake flask culture using HEK SFM medium and in-house feeds were first scaled up to a 3 L bioreactor to test the growth of HEK 293SF cells. In parallel, a control culture taken out from the bioreactor after inoculation was maintained in a shake flask and fed under the same conditions used for the 3 L bioreactor. Figure 5A reveals that the maximum viable cell density was only 9 × 10^6^ cells/mL in the bioreactor culture, much lower than the 16 × 10^6^ cells/mL obtained in the control shake flask culture.

The glucose consumption rate in the bioreactor culture was much faster than the control shake flask culture, almost resulting in a glucose depletion on day 4 (Figure 5B). As a result of the high glucose consumption rate, the concentration of lactate accumulated in the bioreactor culture over the time course was much higher, exemplified by the respective maximum concentration of 38- and 19.6-mM lactate detected in the bioreactor and shake flask culture. Samples taken from the bioreactor and control flask culture over the time course were analyzed for residual concentration of amino acids. However, analytical results revealed no significant difference in the consumption rate of amino acids between the bioreactor and flask cultures. No amino acids were depleted during the time course of culture.

The production of Ad5-GFP was also scaled up to a 3 L bioreactor under the fed-batch conditions. One external shake flask control with culture taken out from the bioreactor after inoculation and one internal control shake flask culture taken out from the bioreactor after the viral infection were conducted parallel to the bioreactor culture. Figure 5C depicts that the virus productivity was 30.4 × 10^9^ vp/mL in the external control culture infected at a cell density of 5.1 × 10^6^ cells/mL, while the titer was only 1.8 and 0.5 × 10^9^ vp/mL, respectively, in the bioreactor and internal control cultures infected at a cell density of 4.1 × 10^6^ cells/mL. The maximum concentration of lactate and ammonia in the bioreactor culture was 20.6 and 2.1 mM, respectively. This result clearly suggests that the virus productivity in the bioreactor and internal cultures was affected by either nutrient limitation, metabolite inhibition or cell status, or a combination of these factors, during cell growth and virus production.

*Cell-specific virus productivity of HEK 293SF culture grown to different cell densities over the time course of fed-batch culture process:* A set of experiments designed according to Figure 1 was carried out to better understand the factor(s) causing the drastic decline in the virus production in the fed-batch bioreactor culture. The productivity of fed-batch cultures grown in 3 L bioreactor and 2 L shake flask to different cell densities was investigated. Figure 6A depicts a decline of cell-specific virus productivity from about 6000 vp/cell in the culture taken from the bioreactor and infected at 1.1 × 10^6^ cells/mL to productivity of <100 vp/cell in the culture from the bioreactor and infected at 3.3 or 6.5 × 10^6^ cells/mL. This represented a more than 98% decrease in cell-specific productivity and was similar to the result obtained in the bioreactor and internal control cultures described in the previous paragraph (Figure 5C). There was a significant correlation between the cell-specific productivity (CSP) and the total cell density (TCD) of culture taken from the 3 L bioreactor before the viral infection (CSP = −1021 × TCD + 5641; R^2^ = 0.65). The cell-specific productivity also declined in the cultures from the 2 L shake flask and infected at higher cell densities. The percent reduction in specific productivity increased with the increasing cell density at infection: 65%, 74% and 94% for the cultures infected at 3.1 × 10^6^, 4.2 × 10^6^ and 7.3 × 10^6^ cells/mL, respectively, in comparison to the cell-specific productivity obtained in a culture infected at 1.2 × 10^6^ cells/mL. There was also a strong correlation between the CSP and TCD (CSP = −892 × TCD + 6032; R^2^ = 0.81).

Data in Figure 6B show that when the cultures harvested at different cell densities from the 3 L bioreactor and 2 L shake flask were centrifuged and cell pellets were resuspended in fresh HEK SFM medium at 1 × 10^6^ cells/mL before the viral infection, there was a trend of only mild decline in the cell-specific virus productivity with the increasing cell density in the harvested culture. The cell-specific productivity was in the range of 6500 ± 1050 vp/cell when excluding the highest and lowest productivity (9800 and 3400 vp/cell), indicating that the cell-specific productivity was not dramatically affected by the cell density of culture. This trend was reflected by a weak correlation between the CSP and TCD in the data from the 3 L bioreactor culture (CSP = −549 × TCD + 8152; R^2^ = 0.37) or from the 2 L culture (CSP = −439 × TCD + 8786; R^2^ = 0.32). This result might suggest that the status (quality) of cells was not the main factor causing the drastic decline in the virus production when the culture was infected at high cell densities without medium exchange.

The maximum concentration of lactate and ammonia was 39 and 2.8 mM, respectively, in the fed-batch bioreactor culture when the cell density was at 6.5 × 10^6^ cells/mL and was last withdrawn for the virus infection. The increased concentration of these metabolites might contribute to the decline of virus productivity, but seems less likely to be the main contributor to >95% reduction in the virus productivity based on our previous study [19]. All of these data suggested that the declined virus productivity in the bioreactor culture infected at higher cell density might be related to nutrient limitation due to a difference in nutrient consumption rates between the shake flask and bioreactor cultures. Supplementing hydrolysates, such as Sheff-Vax ACF (or ACF in short), provides broad-spectrum nutrients and might alleviate nutrient limitation in the bioreactor culture infected at higher cell density. Therefore, we set out to optimize the nutrient supplies in the culture media and feed to maintain cell-specific productivity.

*Maintaining cell-specific virus productivity in fed-batch bioreactor culture through the improvement of culture media and feed:* HEK SFM medium was fortified with 1 g/L ACF before being used for the bioreactor inoculation. Glucose was included in the first feed to provide an additional 2 g/L glucose (final concentration) in the bioreactor culture to avoid glucose depletion. Figure 7A depicts the growth profile of HEK 293SF cells in 3 L fed-batch bioreactor culture under improved nutritional conditions. The total cell density at infection was 4.8 × 10^6^ cells/mL with a viability of 98%, and reached 6.2 × 10^6^ cells/mL with a viability of 73% at 48 hpi. The glucose was not depleted in the culture during the time course. The maximal molar lactate and ammonia concentrations reached 57 and 2.8 mM, respectively, and the osmolality was 347 mOsm/kg at 48 hpi (Figure 7B).

Volumetric titer of samples taken from the bioreactor at 43 and 48 hpi was 26.3 and 31.3 × 10^9^ vp/mL, respectively, corresponding to a respective cell-specific virus productivity of 5479 and 6521 vp/cell. Volumetric productivity of an external control shake flask culture conducted in parallel and infected at 5.2 × 10^6^ cells/mL was 33.0 × 10^9^ vp/mL (Figure 7C). The cell-specific productivity from this fed-batch bioreactor or shake flask culture was similar to the results obtained in the shake flask or bioreactor cultures infected at 1 × 10^6^ cells/mL without feeding or media replacement before the viral infection (Figure 6A), demonstrating that the cell-specific productivity could be maintained in the fed-batch bioreactor culture when the nutrient supplements in the culture were balanced. This result also demonstrated the feasibility of improving volumetric virus production through infecting cell culture at higher density in fed-batch culture.

## 4. Discussion

With recent advancements in the development of culture media, some commercial media can support the growth of HEK 293SF cells up to around 5 × 10^6^ cells/mL or higher in batch culture. However, the optimal cell density at infection for maximum volumetric virus production is still less than 2 × 10^6^ cells/mL, which was evidenced again in this study. The data in Figure 2 revealed that there is no correlation between the virus productivity and achievable maximum cell density (R^2^ = 0.03) among the 5 cell culture media tested in this study. The observed decrease in the specific productivity at higher cell densities was further exemplified by the 16 × 10^6^ cells/mL obtained in the fed-batch culture using commercial SFM4transfx-293 medium and CB5 feed (Figure 3). In this process, when the culture was infected at 3 or 4 × 10^6^ cells/mL, the volumetric productivity dropped by almost one log, down to 6 × 10^8^ vp/mL from 4.5 × 10^9^ vp/mL obtained in the culture infected at 1 × 10^6^ cells/mL. These results clearly show that a culture medium that supports a high density of cell growth does not necessarily warrant higher productivity. This indicates that, although appropriate cultivation strategies can be employed to increase cell densities in culture, it is often difficult to maintain cell-specific and volumetric productivities. This challenge is more likely due to the biphasic processes of cell culture-derived viral and vector production, in which an initial cell growth phase is followed by a virus replication phase initiated by virus infection [23]. The cell growth phase and virus production phase might have different nutritional requirements.

The development of cell culture processes to increase adenovirus production has been often approached by designing feeding strategies that improve nutrient supply and reduce the accumulation of inhibitory metabolite(s) [7,24]. Since most commercial culture media are based on proprietary formulations, very few data have been published on nutritional requirements during cell growth as well as virus production. The lack of basic information on the composition and the complexity of commercial cell culture media are obstacles to developing nutrient-specific strategies and customized media that can support both high cell density and cell-specific productivity. We have exploited our experience in developing fit-for-purpose cell culture media and employed a rational approach through the design of experiments to develop and customize an in-house serum-free medium (HEK-SFM) and feeds that can support a cell density of 5 × 10^6^ cells/mL in batch culture and up to 16 × 10^6^ viable cells/mL in fed-batch culture (Figure 4A). More importantly, through the optimization of in-house media and feed, the volumetric virus production dramatically improved to 3.0 × 10^10^ vp/mL, and cell-specific virus productivity was maintained in the fed-batch cultures infected at higher cell densities (2.5 and 5 × 10^6^ cells/mL).

Scaling up the fed-batch culture in a 3 L bench scale bioreactor to achieve high cell density and Ad5 productivity was challenging. The maximum viable cell density in the fed-batch bioreactor culture did not exceed 9 × 10^6^ cells/mL, and the volumetric productivity dropped to 1.8 × 10^9^ vp/mL when the fed-batch bioreactor culture was infected at 4.1 × 10^6^ cells/mL, suggesting unfavorable conditions in the bioreactor for the cell growth and virus production. The unfavorable conditions could be attributed to increased concentration of accumulated metabolites (such as lactate and ammonia) and/or depleted nutrients (such as glucose) due to increased consumption rate of nutrients caused by shear stress [25,26] under the bioreactor or even large shake flask conditions. Interestingly, the cell-specific productivity of the cultures (Figure 6B) grown to different cell densities and withdrawn from the 3 L bioreactor revealed that the status of cells in supporting the virus production was not dramatically and negatively impacted when the cells were resuspended to fresh media at a density of 1 × 10^6^ cells/mL and infected.

Nutrient limitation has been frequently cited as the main cause for decreased platform performance at high cell density. Amino acids, glucose and easily-analyzed components in spent media were initially examined. However, aside from glucose, no significant nutrient depletion was measured. Due to the complexity of medium formulation and the limitation of analytical techniques, it is very challenging to quantify all the potential limiting components in spent media. Many trials, where individual or a group of components were supplemented to the cultures grown to 5 × 10^6^ cells/mL and withdrawn from the 3 L bioreactor before the viral infection, resulted in no or insignificant improvements. A significant increase in the virus production was observed and the cell-specific productivity was maintained only when the culture medium (HEK-SFM) was supplemented with 1 g/L of ACF prior to the cell inoculation. This result might suggest that nutrient consumption is complex in the cell culture process, especially in the biphasic cell growth and virus production process. Identification of potential limiting nutrients is challenging, and the increased consumption of key nutrient(s) in the bioreactor culture could not be effectively compensated by supplementing fed-batch cultures with an individual component in order to improve the virus production. A hydrolysate such as ACF provides a much broader range of nutrients and is a fast approach to offset the depleted key nutrient(s). The result from this study indicated that the decline in the virus production in the 3 L bioreactor fed-batch culture infected at higher cell density was indeed at least partially due to nutrient limitation and could be minimized through nutrient supplementation. To our knowledge, the work reported in this study is the first demonstration of high titer Ad5 production in fed-batch bioreactor culture infected at a cell density of up to 5 × 10^6^ cells/mL (highest so far) while maintaining cell-specific virus productivity.

## 5. Conclusions

Overall, this study confirms that a culture medium and/or process, which is superior in supporting the growth of HEK 293SF cells to high density, does not necessarily correlate with high vector productivity. Culture media and/or feeding strategies could be improved to provide optimal nutrients required for cell growth and virus production in cultures infected at higher density in order to increase volumetric virus productivity. Further optimization of culture media and feed might be required when high cell density culture is scaled up to the bioreactor to achieve high-titer virus production. Fed-batch culture could be a simple and cost-effective process for high titer production of viruses and viral vectors through infecting culture at high density while maintaining cell-specific virus productivity.

## Figures and Tables

**Figure 1 vaccines-12-00524-f001:**
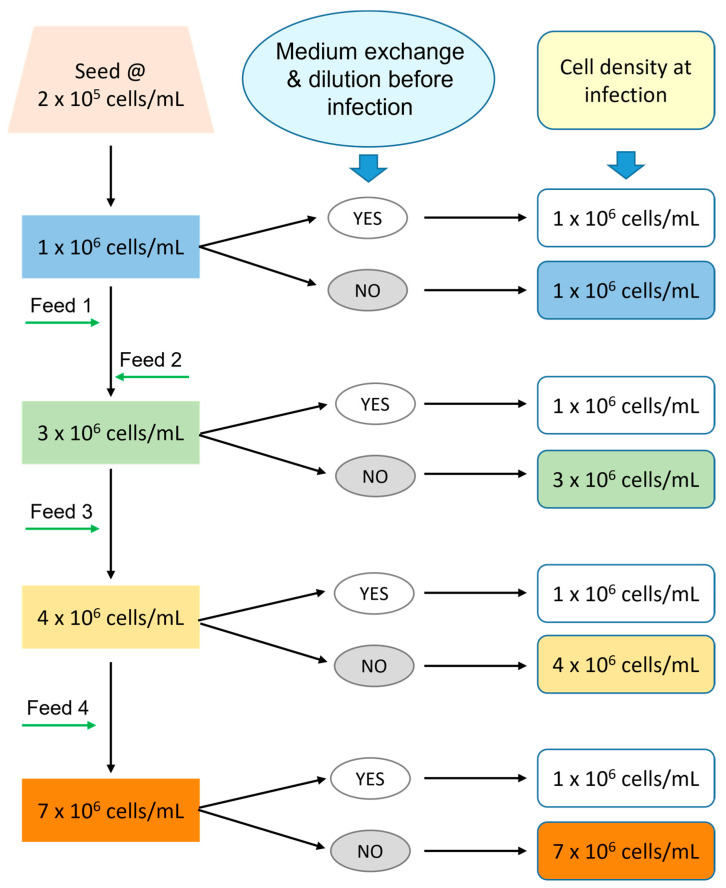
Scheme of the experiment designed to assess the influence of cell status (quality) on production of Ad5-GFP. HEK 293SF cells were cultivated in 2 L shake flask and 3 L bioreactor, respectively, with HEK SFM medium, and fed with the in-house developed feeds. When the cultures reached different cell concentrations, a portion of the culture was withdrawn and infected or centrifuged and then resuspended to fresh media at 1 × 10^6^ cells/mL (medium exchange and dilution) before infection.

**Figure 2 vaccines-12-00524-f002:**
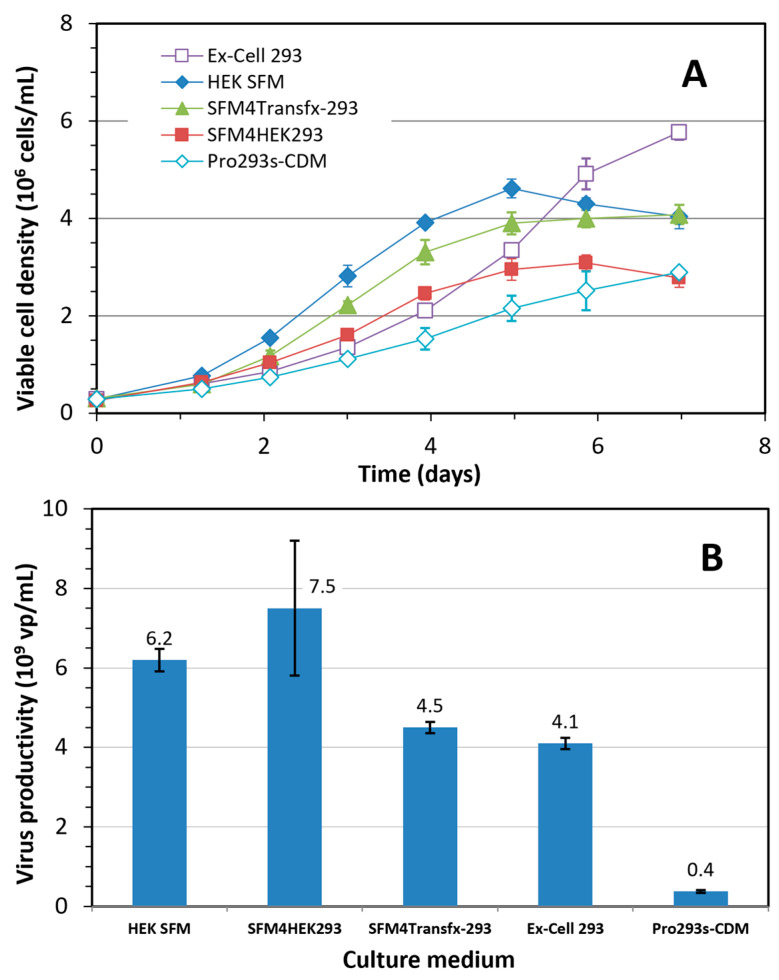
(**A**) Growth of HEK 293SF cells in batch cultures cultivated in four commercial and one in-house-developed serum-free media (HEK SFM); (**B**) Volumetric productivity of Ad5-GFP in batch cultures grown in the above five media to 1 × 10^6^ cells/mL and then infected with Ad5-GFP at an MOI of 10. Mean and standard deviation of two independent cultivations.

**Figure 3 vaccines-12-00524-f003:**
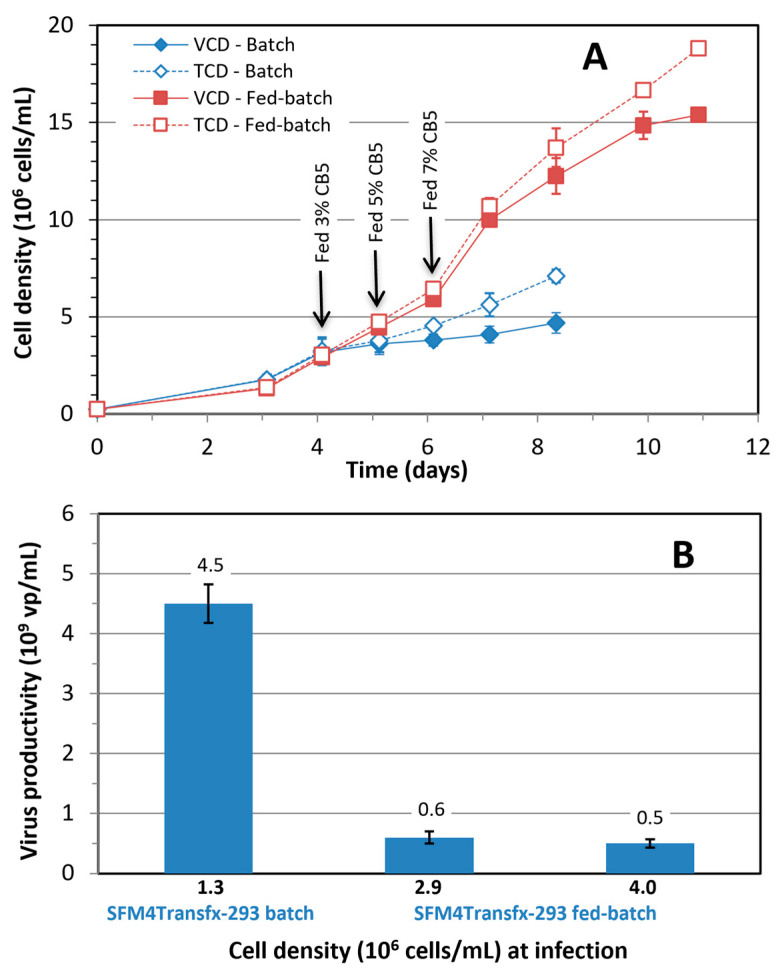
(**A**) Growth of HEK 293SF cells in fed-batch culture cultivated in SFM4Transfx-293 medium and fed with CB5; (**B**) Comparison of volumetric productivity of Ad5-GFP in batch cultivated in SFM4Transfx-293 medium and fed-batch cultures fed with CB5, and infected at three different cell densities. Mean and standard deviation of two independent cultivations.

**Figure 4 vaccines-12-00524-f004:**
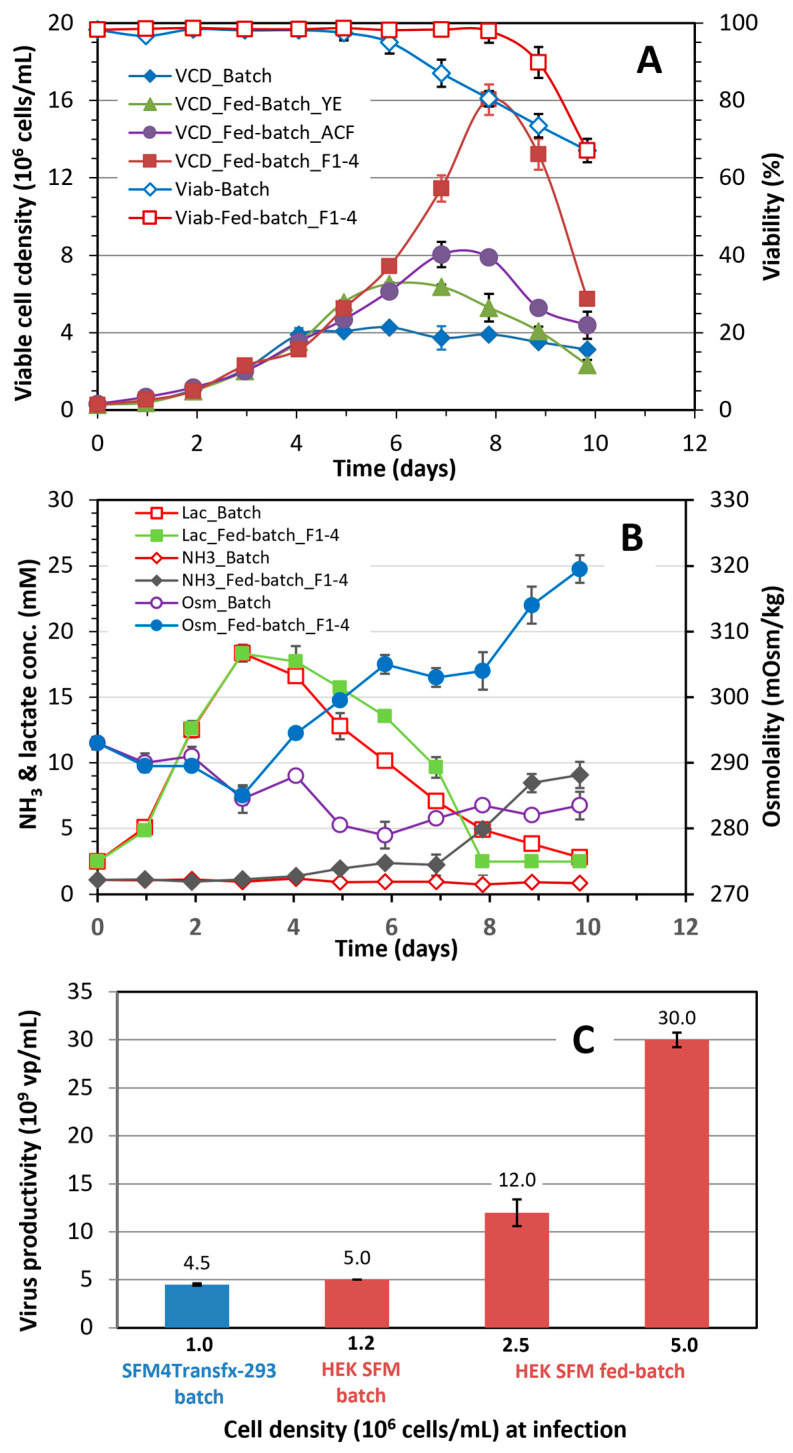
(**A**) Growth profile of HEK 293SF cells in batch culture (◆) cultivated in HEK SFM media, fed-batch cultures fed with 1g/L yeast extract (▲), 1g/L ACF (●) or with in-house developed feeds (F1-4, ■) at various times; and cell viability of the batch (◇) and fed-batch (☐) culture with feed (F1-4). Cell viability for Fed-batch_YE and Fed-batch_ACF were not shown; (**B**) Lactate (Lac) and ammonia (NH_3_) concentration, and osmolality in the batch and fed-batch cultures; (**C**) Comparison of volumetric productivity of Ad5-GFP in batch and fed-batch cultures infected at three different cell densities. Mean and standard deviation of two independent cultivations.

**Figure 5 vaccines-12-00524-f005:**
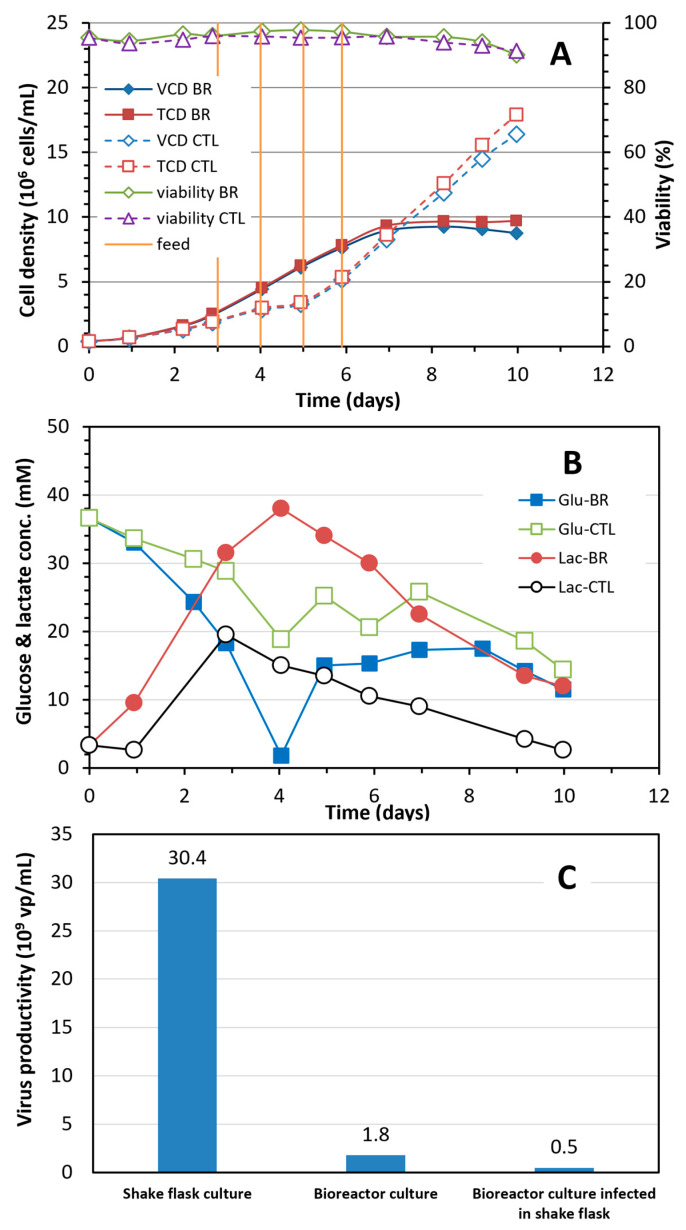
(**A**) Comparison of HEK 293SF cell growth in 3 L fed-batch bioreactor (BR) and control shake flask (CTL) culture during the process scale-up. (**B**) Glucose (Glu) consumption and lactate (Lac) production in the 3 L bioreactor and shake flask culture. There was no significant difference in the ammonia concentration. (**C**) Volumetric virus productivity in a fed-batch culture grown in a shake flask infected at 5.1 × 10^6^ cells/mL and bioreactor culture infected at 4.1 × 10^6^ cells/mL, and in a shake flask with culture taken from the bioreactor immediately after infection.

**Figure 6 vaccines-12-00524-f006:**
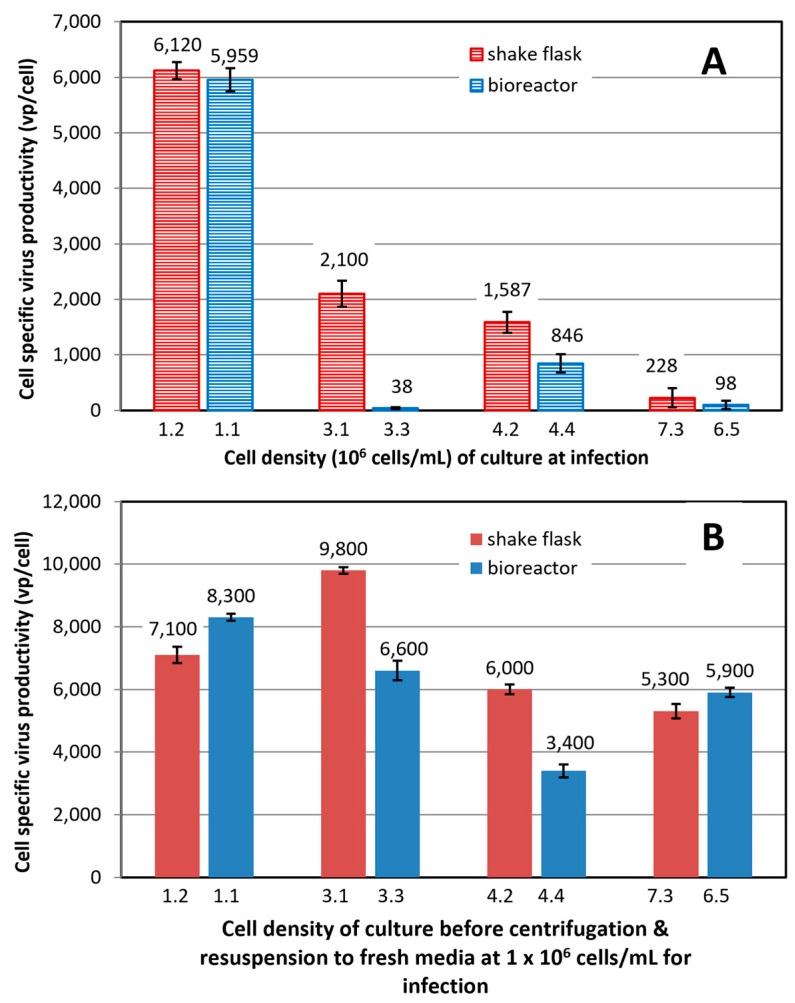
(**A**) Cell-specific productivity of cultures taken from a 2 L shake flask (with 700 mL working culture volume) and 3 L bioreactor fed-batch cultures grown to different cell densities, aliquoted to 125 mL shake flask at 25 mL each, and then infected with Ad5-GFP at MOI of 10; (**B**) Cell-specific productivity of cultures taken from the 2 L shake flask and 3 L bioreactor, and centrifuged; Cell pellet was suspended in fresh HEK SFM medium at 1 × 10^6^ cells/mL, and then infected. Mean and standard deviation of two independent cultivations.

**Figure 7 vaccines-12-00524-f007:**
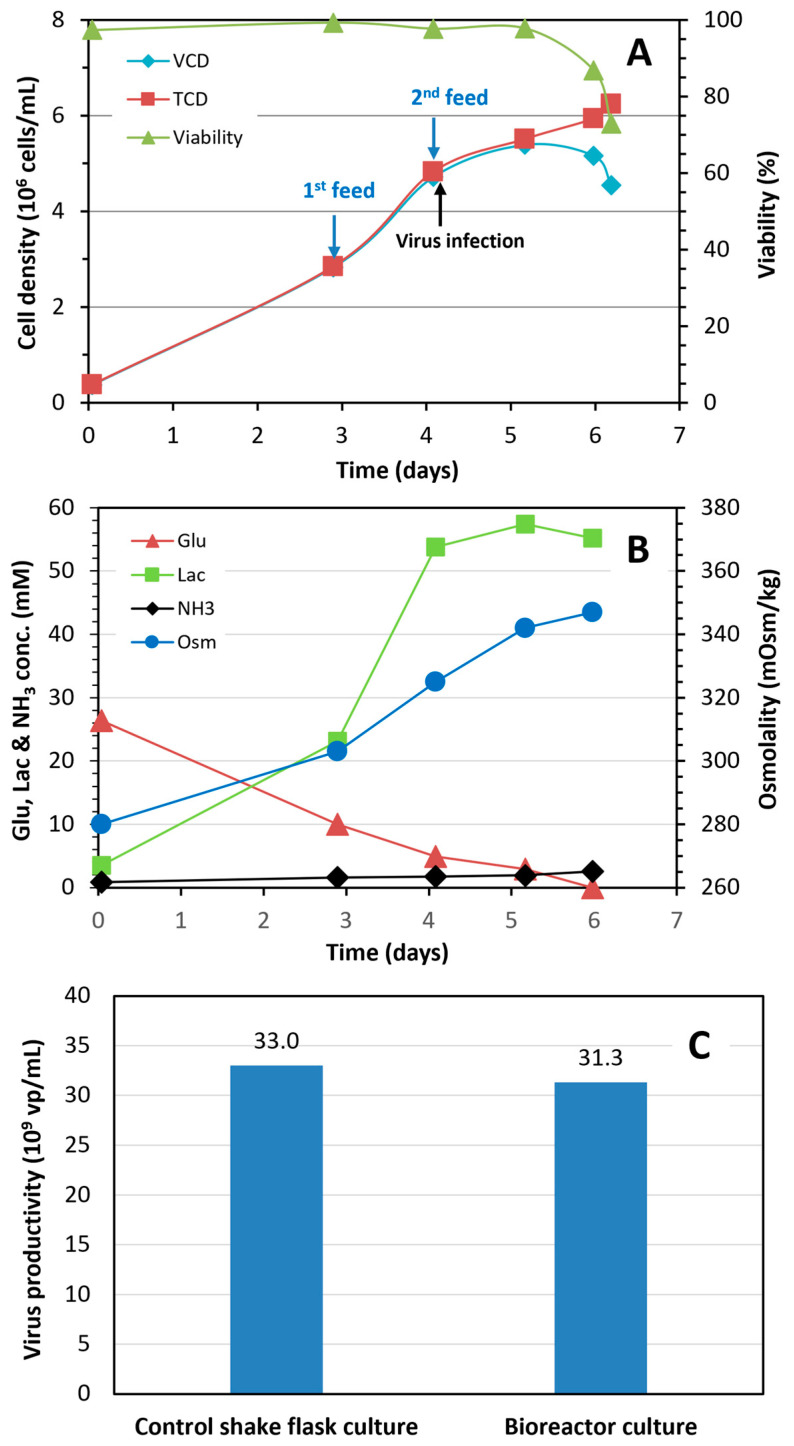
(**A**) Cell growth profile in a 3 L fed-batch bioreactor culture conducted under optimized feeding conditions for production of Ad5-GFP; (**B**) Culture osmolality and concentration of glucose, lactate and ammonia; (**C**) Volumetric virus productivity in control shake flask fed-batch culture infected at 5.2 × 10^6^ cells/mL and bioreactor fed-batch culture infected at 4.8 × 10^6^ cells/mL.

**Table 1 vaccines-12-00524-t001:** Composition of in-house developed feeds and feeding schedule.

Feeds	Components	Final Concentration in Culture (mg/L)
Feed 1 (Day 3)		
	Yeast extract	1000
	Sheff-Vax ACF	1000
Feed 2 (Day 4)		
	Glucose	2000
	yeast extract	1000
	L-aspartic acid	145
	L-serine	466
	L-asparagine	560
	L-arginine	312
	L-cysteine.HCl	58
	L-valine	228
	L-methionine	67
	L-isoleucine	277
	L-leucine	400
	Adenine sulfate	5
	Adenosine 5′-triphosphate.2Na (ATP)	0.0015
	Adenylic acid	0.0003
	Thymine	0.0004
	Thymidine	0.0006
	Guanine.HCl	0.0005
	Hypoxanthine	0.002
	Uracil	0.00045
	Xanthine	0.00045
Feed 3 (Day 5)		
	Yeast extract	2000
Feed 4 (Day 6)		
	Yeast extract	1000

## Data Availability

No new data were created or analyzed in this study. Data sharing is not applicable to this article.

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
