# Peer review of "Optimization of Culture Media and Feeding Strategy for High Titer Production of an Adenoviral Vector in HEK 293 Fed-Batch Culture"

_vaccines, 2024, doi:10.3390/vaccines12050524_

Round 1
Reviewer 1 Report
Comments and Suggestions for Authors
Overall comments: Propagation of adenovirus vectors to high yields is important for production of adenovirus vectors as vaccines and for gene therapy. Typically, suspension cultures are used for this purpose. It seems logical to think that being able to amplify cells to high density in suspension culture should lead to production of high virus yields using the same culture conditions. The authors report here that this assumption is not true, that conditions supporting amplification of cells to high density can compromise the yield of virus from these cells. It is important that culture conditions have to be optimized for both cell amplification and virus production. Given the practical relevance of this work to the large-scale production of adenovirus vectors, I feel that the results should be available to people working in this field.
There are some specific comments about wording that need attention; these comments are listed below.
Specific comments:
Line 141 – amino acids
Lines 190, 204, 206 – delete “respectively”. It’s awkward and not necessary.
Line 193 – cells
Line 195 – cells’
- latter
Line 213 – Growth
Fig 3A – it’s not obvious what VCC and TCC are – the terms should be written out in the Fig legend. Also, it’s not clear what the three arrows signify. That too should be in the Fig legend.
Line 221 – “very promising” not “a very promising”
Line 238 – delete “respectively”
Line 239 – add “respectively” at the end of the sentence ie “”…. (Fed-batch _ ACF), respectively.”
Line 251 – delete “of”
Line 252 – infected at 1.2 x 106 cells/ml
Line 257 – delete respectively
Fig 4B – it’s not clear what the blue and purple circles are
Line 300 – experiments
Lines 311, 312 – delete “respectively”
Line 312 – delete “of”
Line 312 – add “respectively” after 7.3 x 106 cells /ml
- delete “to the”
Line 378 – show
Line 396 – experiments
Line 428 – “an individual component”
Line 430 – Add period after “virus production”. Start new sentence with “A hydrolysate …..”
Line 436 – “warrant” isn’t the right word here. Suggestions “guarantee” or “correlate with”
Comments on the Quality of English LanguageThe language, in general, is fine. There are a few places that need minor changes and those have been identified in the "Comments to Authors" section.
Author Response
Thank you very much for your valuable suggestions and comments on our manuscript. Those comments are of great assistance to us for improving and revising our manuscript.
Response to the reviewer’s comments: All specific comments regarding wording have been addressed in the revised manuscript. The section of materials and methods has been revised for readability. The corresponding revisions/corrections highlighted in track changes in the re-submitted files.
Reviewer 2 Report
Comments and Suggestions for Authors
Adenoviral-based vectors have been extensively developed for vaccines and gene therapy, and thus the scale production of Ad-based vectors is a critical issue. In this manuscript, authors reported the optimized strategy for high titer production of Ad vector. Basically, it is of considerable interest, but there are some concerns regarding this manuscript.
1. Mainly described a method to manufacturing recombinant Ad, and there was no any data about the validation of its immunogenicity and vaccinology. Whether this is matched to scope of Vaccines journals?
2. The format of the Figures does not meet the requirements for a published paper.
Reviewer 3 Report
Comments and Suggestions for Authors
Title: Optimization of culture media and feeding strategy for high titer production of an adenoviral vector in HEK 293 fed-batch culture
The manuscript aims to determine optimal culture conditions for suspension HEK293 cells for adenovirus propagation. The authors show that addition of nutrients to the culture during cell growth can be used to increase doubling time and maximum growth concentration. The authors demonstrate that cell density limits virus propagation. The authors show that bioreactors require more nutrients than shaker flask culture. The authors demonstrate a method of fed-batch culture supplementing with ACF to increase adenovirus production. Overall, the manuscript provides in-house media recipe and protocol for HEK293 cell culture and Ad5 production in a bioreactor.
· The major weakness of the manuscript is lack of clarity, it was difficult to follow from the text and figure legends exact how the experiment was performed. Increased description of experimental design or schematics would help the readability of the manuscript.
· Error bars are not always present and the number of replicates not always stated.
· No statistical comparison performed to support conclusions
· Figure 2 – what day was virus infection done? They were all infected at same density, were cells infected as soon as they reached that density?
· Correlations between doubling times and virus productivity could help.
· Figure 3, what do arrows correspond to? Fed times with Cb5? Define VCC and TCC.
· “respectively” is often used and it is difficult follow
· Figure 4B, what is blue line?
· Figure 5 – no error bars or replicates mentioned.
· Figure 5 and Figure 7, virus data not graphed and no replicates or standard deviation mentioned.
Comments on the Quality of English Language
Quality of English is fine.
Round 2
Reviewer 2 Report
Comments and Suggestions for Authors
This version has been greatly improved, and I think it is suitable for publication.